# Unconventional Hund metal in a weak itinerant ferromagnet

Xiang Chen[1], Igor Krivenko [2], Matthew B. Stone [3], Alexander I. Kolesnikov [3], Thomas Wolf[4], Dmitry Reznik [5], Kevin S. Bedell[6], Frank Lechermann[7✉] & Stephen D. Wilson [1✉]

The physics of weak itinerant ferromagnets is challenging due to their small magnetic moments and the ambiguous role of local interactions governing their electronic properties, many of which violate Fermi-liquid theory. While magnetic fluctuations play an important role in the materials' unusual electronic states, the nature of these fluctuations and the paradigms through which they arise remain debated. Here we use inelastic neutron scattering to study magnetic fluctuations in the canonical weak itinerant ferromagnet MnSi. Data reveal that short-wavelength magnons continue to propagate until a mode crossing predicted for strongly interacting quasiparticles is reached, and the local susceptibility peaks at a coherence energy predicted for a correlated Hund metal by first-principles many-body theory. Scattering between electrons and orbital and spin fluctuations in MnSi can be understood at the local level to generate its non-Fermi liquid character. These results provide crucial insight into the role of interorbital Hund's exchange within the broader class of enigmatic multiband itinerant, weak ferromagnets.

[1] Materials Department, University of California, Santa Barbara, CA 93106, USA. [2] Department of Physics, University of Michigan, Ann Arbor, MI 48109, USA. [3] Neutron Scattering Division, Oak Ridge National Laboratory, Oak Ridge, TN 37831, USA. [4] Institute for Solid State Physics, Karlsruhe Institute of Technology, 76131 Karlsruhe, Germany. [5] Department of Physics, University of Colorado at Boulder, Boulder, CO 80309, USA. [6] Department of Physics, Boston College, Chestnut Hill, MA 02467, USA. [7] I. Institut für Theoretische Physik, Universität Hamburg, 20355 Hamburg, Germany. ✉email: frank.lechermann@physnet.uni-hamburg.de; stephendwilson@ucsb.edu

Understanding the underlying interactions in weak itinerant ferromagnets (e.g. $ZrZn_2$, MnSi or $Ni_3Al$) poses a longstanding challenge in condensed matter physics[1,2]. The seminal theory of itinerant spin fluctuations by Moriya[3] provides important insight, yet neglects the effects of local and short-range effects in their complex electronic properties. Clear indications of these strong correlation effects such as the considerable enhancement of the low-temperature electronic specific-heat and non-Fermi-liquid (NFL) behavior have posed a persistent challenge to synthesizing a complete understanding of both the many-body phenomenology and the (local) electronic structure of these transition-metal compounds. Common, highly local, approaches built from Mott-physics usually associated with various transition-metal oxides also fail in this endeavor.

The phase behaviors found in MnSi are particularly emblematic of the complexity of the interplay between local interactions and itinerancy at this frontier, and the coupling between itinerant electrons and magnetic order in MnSi has remained of sustained interest for decades[4–9]. Local magnetic interactions favor a ferromagnetic phase at low temperature; however its non-centrosymmetric lattice allows a global Dzyaloshinskii–Moriya interaction to create a long period, helical modulation of the locally ferromagnetic moments[10]. As such, MnSi has served as a model material at both the long-wavelength limit for exploring topological spin defects[6] (skyrmions)[11] coupling to charge carriers[12,13] as well at the short-wavelength limit, where weak ferromagnetism provides a rare experimental realization of magnon decay within a particle–hole (Stoner) continuum[14]. Underlying this is the notion of prototypical itinerancy in MnSi's charge and spin degrees of freedom—a view supported by the observation of strongly screened moments[15,16], and the success of conventional band-based models in capturing the qualitative features of low-energy magnon propagation[11,17].

MnSi nevertheless exhibits signs of significant electronic correlations. Large electron mass renormalization and strong quasiparticle damping have been detected in both de Haas-van-

Alphen and angle-resolved photoemission (ARPES) measurements[18–20]. An extended NFL phase has been identified under pressure[4], and indications for NFL behavior are also observed in the optical conductivity[21] at ambient conditions. The root of these departures from the expectations of conventional Fermi-liquid theory and the role of electron correlations remain open questions.

The interplay of the charge and spin degrees of freedom in itinerant magnets is effectively studied via measurements of the dynamic magnetic susceptibility $\chi''(\mathbf{q},E)$[22,23]. Inelastic neutron scattering measurements mapping broad regions of momentum ($\mathbf{q}$) and energy ($E$) space directly sample $\chi''(\mathbf{q},E)$, and, when combined with density functional theory (DFT) paired with dynamical mean-field theory (DMFT) to describe realistic correlation effects, considerable insight can be gained. For instance, DFT+DMFT can probe the energy scales of local orbital and local spin coherence directly accessible in neutron measurements[24,25]. This aspect is critical for identifying and understanding properties of Hund metals[26–28].

In a Hund metal, strong electronic correlations do not stem from the intraorbital Coulomb repulsion $U$, but instead mainly arise from the effects of the interorbital Hund's exchange $J_H$ acting within a broad bandwidth orbital manifold—optimally one with an occupation of one additional electron or hole away from half filling[29]. The separation of spin and orbital screening energy scales is a common feature of these unusual metals, with Fermi-liquid behavior only appearing well below the lowest screening scale. In the separation regime, incoherent phenomenology appears such as bad-metal or non-Fermi-liquid behavior[26,29], and prominent examples of Hund metals are established in iron-based superconductors and in ruthenates[29–31].

In this work, we present a study of the MnSi high-frequency dynamic susceptibility $\chi''(\mathbf{q}, E)$ unveiling signatures of strong correlation effects consistent with those of a Hund metal. Over-damped low-frequency magnon modes disperse within the particle–hole continuum until an anomalous, intermediate energy

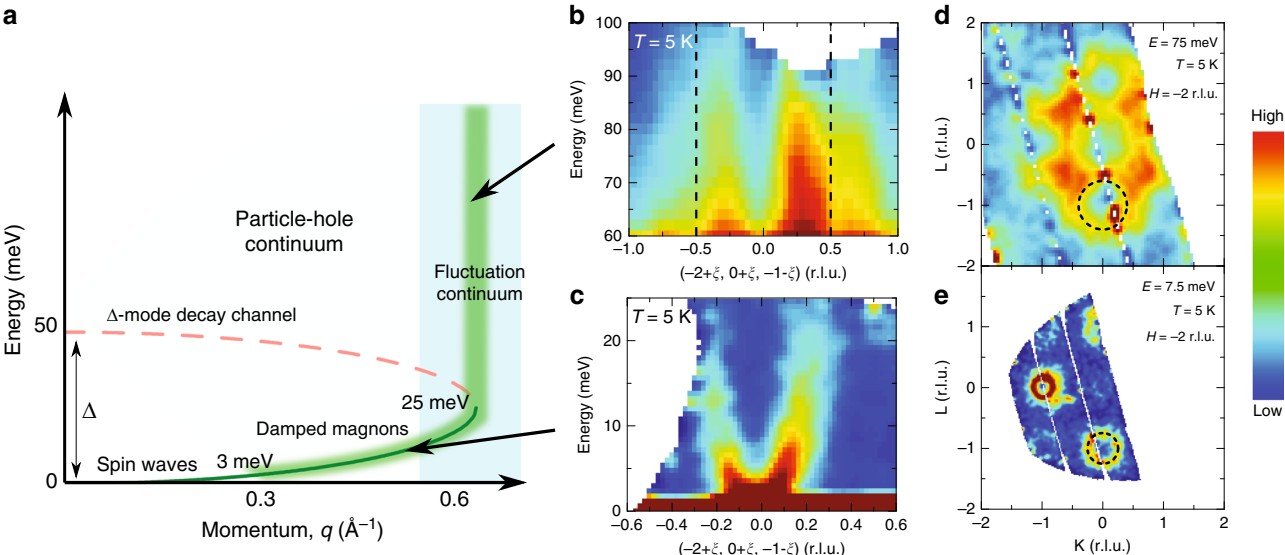

**Fig. 1 Dispersion of spin fluctuations in MnSi away from the low-q limit. a** Illustration depicting low-**q** spin waves (green lines) entering the Stoner continuum (shaded light blue) and continuing dispersing toward the zone boundary. A second decay channel is accessed at the crossing with the Δ-mode (dashed red line) and a continuum of spin fluctuations onsets (vertical shaded green bar). **b** High-energy continuum of spin excitations in MnSi collected at $E_i = 150$ meV and $T = 5$ K. **c** Low-energy dispersion of damped magnons within the Stoner continuum, collected with $E_i = 30$ meV at $T = 5$ K. **d, e** The dispersion of spin excitations via constant energy slices of $S(\mathbf{q},E)$ at 75 and 7.5 meV, respectively ($H$ is centered about $H = -2$). Black dashed lines in **b** denote the zone boundaries at $\xi = 0.5$ r.l.u. Black dashed circles in **d** and **e** signal the ring-like spin wave excitation, where the radii are in line with the fitted values from **b** and **c** and Fig. 2. The color bar denotes scattering intensity in arbitrary units with high scattering intensity in red and low scattering intensity in blue.

scale is reached and the modes' coherence are abruptly lost into a non-dispersive continuum of excitations. This energy scale is captured within DFT+DMFT calculations incorporating both $J_H$ and a modest Hubbard $U$, where the momentum-integrated local orbital susceptibility $\chi''_{\mathrm{loc}}(E)$ peaks at the onset of the magnon continuum. Our results capture the previously reported violations of Fermi-liquid behavior in MnSi as seen by optics[21] and demonstrate that correlations originating from $J_H$ drive substantial renormalization of the electronic spectrum and low-energy properties. This establishes MnSi as an example of an unconventional Hund metal, where a lower-lying orbital coherence scale cuts off damped-magnon propagation, and suggests an application of the Hund-metal framework to a broader class of anomalous, weak itinerant ferromagnets[32,33].

## Results

**Neutron scattering experiments.** Well-defined spin waves at low energies are known to appear in the ferromagnetic phase of MnSi[11]; however, the spin excitations at higher energies relevant for understanding interactions within the metallic state are relatively unexplored. This is largely due to the fact that, as the spin waves disperse upward in energy, they enter the Stoner continuum, access a decay channel into the particle–hole excitations above $E \approx 3$ meV ($q > 0.2$ Å$^{-1}$), and rapidly become damped[34]. Once in the Stoner continuum, these damped magnons continue dispersing toward the zone boundary with a similar spin stiffness as that observed in the low-$q$ regime—an overall phenomenology consistent with a number of itinerant magnets[35–38]. Figure 1a illustrates this progression of magnetic excitations as a function increasing crystal momentum.

Our inelastic neutron scattering measurements begin by exploring spin excitations within the damped regime where the spin waves have already entered the Stoner continuum. At these $E$ and $q$ values, the low-energy/long-wavelength helical modulation of the ferromagnetic state can be neglected, and the system may be viewed as damped ferromagnetic spin waves. Figure 1b–e illustrates the dispersion of these excitations about the $\mathbf{q} = (-2, 0, -1)$ magnetic zone center both upon entering the Stoner continuum above 3 meV and their transformation into a continuum of non-dispersive fluctuations at higher energies. Notably, this high-energy continuum develops before the magnetic zone boundary is reached at $\xi = 0.5$ r.l.u., which is further illustrated in Fig. 2.

Figure 2a, b shows the parameterization of the spin fluctuations as they disperse within the Stoner continuum of MnSi. Excitations along high symmetry directions are shown as they broaden in $\mathbf{q}$, and the modes' lifetimes decrease. This is parameterized by fitting the excitations to the form $I(E) \propto \frac{\gamma E}{\left(E^2 - E_0^2\right)^2 + (\gamma E)^2}$, where $E_0$ is the mode energy and $\gamma$ plotted in Fig. 2d parameterizes the damping of the mode. As the damped magnons enter the continuum, $\gamma$ increases linearly with increasing energy, consistent with models of coupled localized spins and conduction electrons[39]. At the lowest energy measured ($E = 3$ meV), the spectrometer's resolution may convolve broadening into the $\gamma$-term due to helimagnon modes from multiple allowed domains[17]. However, this effect is quickly superseded as the dynamics cross the threshold into the Stoner continuum ($E > 3$ meV) and become intrinsically damped[14]. Near 30 meV, the dispersion toward the zone boundary ceases and the values of $\gamma$ diverge. This marks the onset of a true continuum of magnetic fluctuations between $\mathbf{q} = 0.6 - 0.75$ Å$^{-1}$ depending on the position within the zone. At all points however, the continuum develops before the wave-vector of the magnetic zone boundary (marked as dashed lines) in Fig. 2a, b. This abrupt decay of spin

fluctuations suggests an additional decay channel is opened inside the continuum.

An added decay channel is predicted when quasiparticle interactions are considered. The inclusion of higher-order Landau parameters developed for an interacting ferromagnetic Fermi liquid predicts the stabilization of a gapped mode comprised of transverse spin fluctuations[40,41] (which we label as the Δ-mode). Although the Δ-mode is difficult to measure directly, it can propagate (albeit damped) through the continuum, and when its dispersion crosses with that of the transverse spin fluctuations (shown in Supplementary Information), an additional damping channel is accessed and the damping rates of the excitations diverge. Model dispersion and lifetimes of these modes in an isotropic model using the $l = 0$ and $l = 1$ Landau parameters for MnSi are plotted in Fig. 2c. The inclusion of the $l = 1$ Fermi-liquid parameter is the source of the many-body spin–orbit interaction in the ferromagnetic phase that stabilizes the Δ-mode[42]. This feature also implies that the correlations go beyond local correlations as in a local model the $l = 1$ parameter is zero[43]. Despite ignoring the complexity of the multiband nature of MnSi and the naive breakdown of the small-$\mathbf{q}$ approximations of the theory as the zone boundary is approached, this picture qualitatively captures the onset of the continuum of spin fluctuations in the data. It also hints that strong quasiparticle correlations are essential to understanding the electronic spectrum of MnSi and invites the exploration of local correlation effects in a multiband picture.

**DFT + DMFT modeling and analysis.** The DFT+DMFT scheme was employed to understand the correlated electronic structure for MnSi's predominant Mn($3d^6$) character as identified experimentally[16]. Figure 3a displays the initial nonmagnetic band structure from DFT. Upon the inclusion of local Coulomb interactions via $U = 2$ eV and $J_H = 0.65$ eV on the Mn sites, the multi-sheet Fermi surface is strongly modified. The spectral function $A(\mathbf{k}, \omega)$ of paramagnetic MnSi (Fig. 3b) acquires substantial band narrowing, sizable shifts of quasiparticle-like dispersion as well as strong incoherence effects away from $\varepsilon_F$. These features are consistent with strong correlation effects identified in de Haas-van-Alphen and ARPES measurements[15,17], where, for instance, the emerging electron pocket at the M point as well as the strongly renormalized occupied states at the Γ point (both absent in DFT) appear. A comparison of the Fermi surfaces computed with DFT versus DFT+DMFT is further shown in Supplementary Fig. 7.

Strong correlation effects resulting from moderate interaction strengths for a $d^6$ transition-metal compound suggest Hund-metal physics as a key driving force. To investigate this further, violations of Fermi-liquid theory—a salient feature of strong Hund metals[24,29]—were sought in the Mn self-energy term $\Sigma_{\mathrm{Mn}}$. By inspecting $\Sigma_{\mathrm{Mn}}(i\omega_n)$ in Matsubara-frequency space and fitting its imaginary part to $F(\omega_n) = K\omega_n^\alpha$, the exponent $\alpha \approx 0.5$–0.6 indeed markedly deviates from the Fermi-liquid value $\alpha_{\mathrm{FL}} = 1$ (Fig. 3d). NFL character surprisingly appears with the moderate value $U = 2$ eV, and the $\{e_g, e_{g'}\}$ orbitals foster stronger correlations than the $a_{1g}$ orbital. Increasing the Hund's exchange $J_H$ not only enhances correlation strength and NFL character, but also increases the orbital-resolved self-energy $\{e_g, e_{g'}\}$ vs. $a_{1g}$ differentiation. This role of $J_H$ as an 'orbital decoupler' is a further indication of the relevance of Hund-metal type interactions in MnSi[32].

From experiments, the low-temperature saturated moment of MnSi is measured to be 0.4 $\mu_B$. At 300 K, DFT+DMFT calculations yield a paramagnetic Mn($3d$) moment value of 0.3 $\mu_B$, which differs from the larger reported local moment of 2.2 $\mu_B$

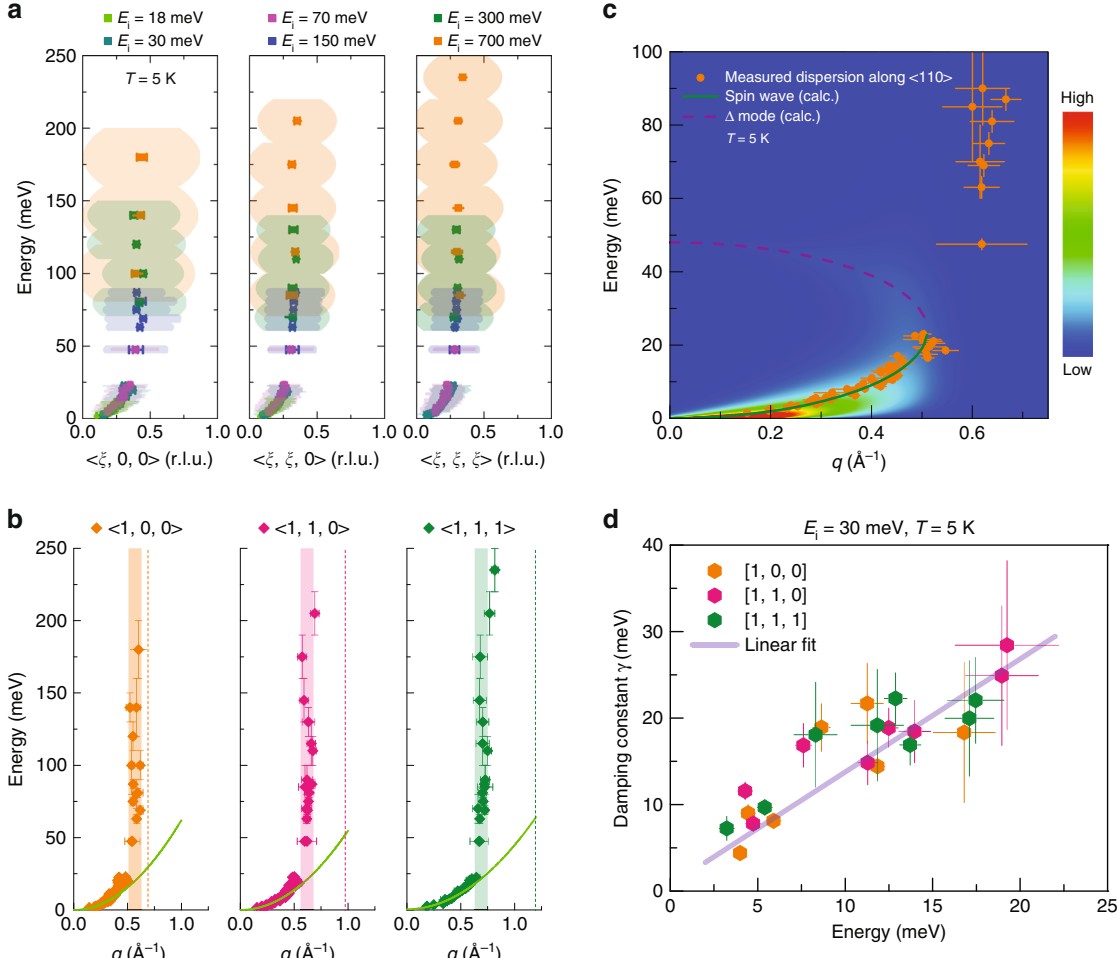

**Fig. 2 Parameterization of high-energy spin excitations in MnSi. a, b** Combined spin wave dispersion relation along high symmetry directions, collected with different incident energies. Shaded regions in **a** are full-width at half maxima (FWHM) from fits to constant $E$ cuts of the spin wave dispersion, demonstrating broad magnetic excitations along both $E$ and **q** directions. Green lines in **b** are fits to the spin wave dispersion in the small **q** limit, showing the $q^2$ dependence of spin wave excitations. Dashed lines in **b** indicate the Brillouin zone boundaries and vertical color bars illustrate the onset of the continuum. **c** Simulation of the expected lifetime broadening of spin excitations as they disperse within the continuum compared with the experimental data along the <110> direction. The color bar denotes scattering intensity in arbitrary units with high intensity in red and low scattering intensity in blue. **d** Damping constants $\gamma$ from damped harmonic oscillator fits to constant **q** cuts of spin wave dispersion below 20 meV, collected with $E_i = 30$ meV at $T = 5$ K. Solid line is a linear fit to the data. Error bars represent one standard deviation of the data.

obtained via Curie–Weiss analysis of the high temperature susceptibility[44,45]. This supports the arguments of Moriya and Kawabata[46], who demonstrated that the true local-moment response can be obscured via long-range spin fluctuations near the zone center, producing a tail in the magnetic susceptibility well above $T_c$. As **q**-dependent spin fluctuations are not included in single-site DMFT, the temperature dependence of the local magnetic moment is very weak, and instead our theoretical result describes the screened long-time moment. Instead, computing the instantaneous moment however renders a larger $m_{Mn}^{in} = 2.73\,\mu_B$. This strong deviation between long- and short-time moment is another fingerprint of Hund metals[25].

The imaginary part of the local (momentum-integrated) magnetic susceptibility $\chi_{loc}''(E)$ can then be analyzed and compared with the experimental results. Figure 4a shows the measured $\chi_{loc}''(E)$ with excitations resolved up to 250 meV. A prominent peak appears near 30 meV near the onset of the continuum of spin fluctuations and suggests a low-frequency loss of spin coherence on the local level. In line with this, our calculations unveil intricate features in the frequency-dependent Mn local orbital and local spin susceptibility, as displayed in

Fig. 4b. Below 1 eV, the spin susceptibility in Fig. 4b shows first a broad maximum at around 320 meV, whereas the orbital susceptibility first peaks at about twice this value. These energy scales may be associated with the preformation of local moments without yet becoming coherently embedded in the itinerant background. The latter process, surprisingly, takes place at much lower energies $E_s = 55$ meV and $E_o = 38$ meV for the spin and the orbital degrees of freedom, respectively. The splitting of these two coherence scales in energy is a key feature of Hund metals[33]. How these processes evolve with changing the $J_H$ value is described in Supplementary Fig. 10 and Note 6.

## Discussion

The observation of coherence scales well below 100 meV with a modest $U$ and the loss of coherence $E_o < E_s$ in MnSi is unusual. Theoretical calculations require vertex corrections to capture these features (Supplementary Fig. 10), whereas the data provide a sharp observation of a low-energy peak in the local spin susceptibility. We note here that experimentally resolving the multipeak structure predicted at low energies is precluded by phonon contamination in this energy regime. At higher energies

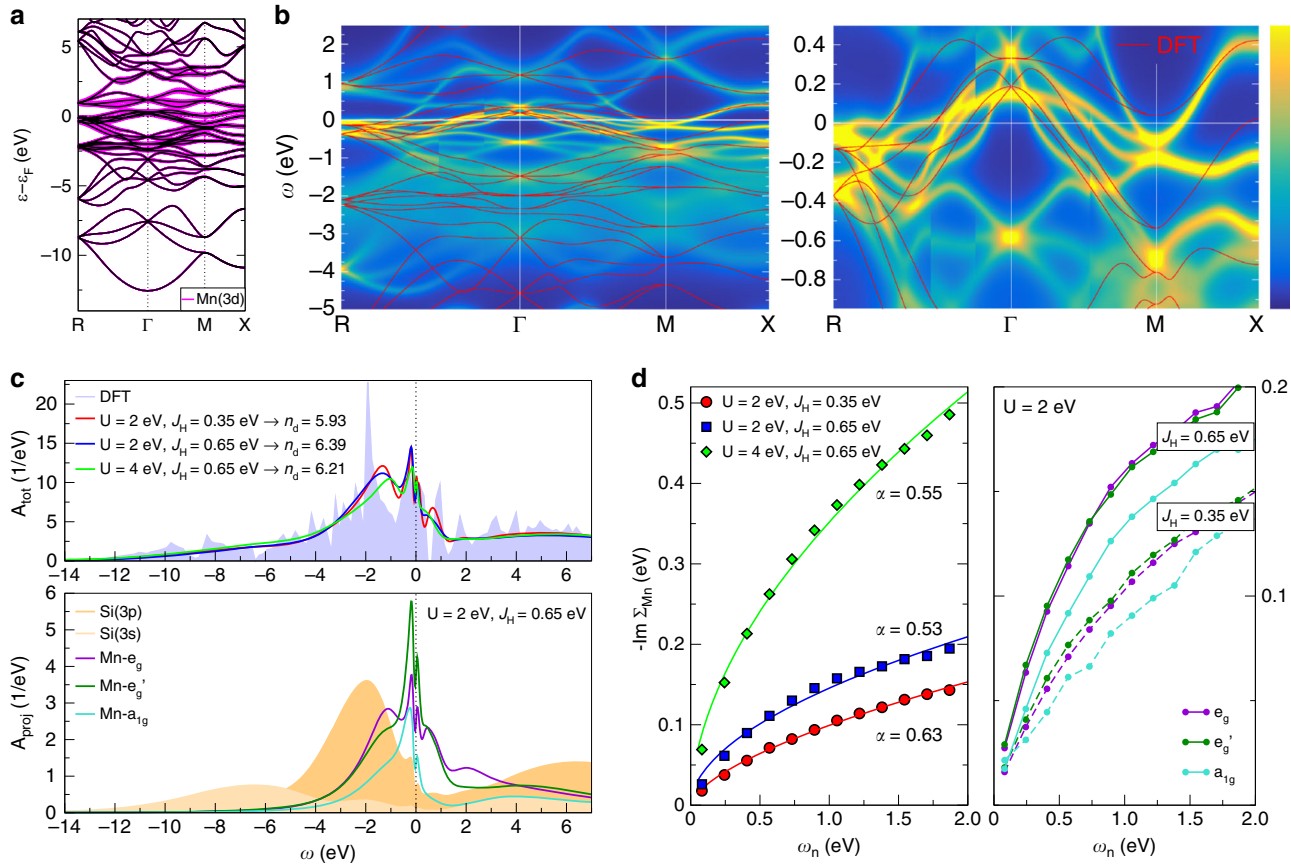

**Fig. 3 One-particle properties of the MnSi correlated electronic structure. a** Band structure from nonmagnetic DFT. Purple color marks the Mn(3$d$) character. **b** DFT+DMFT **k**-resolved spectral function in paramagnetic phase for $U = 2$ eV, $J_H = 0.65$ eV in smaller window (left) and close to the Fermi level (right). High spectral density is noted as yellow in the color bar and low density as dark blue. The nonmagnetic DFT bands are overlayed (red lines). **c** k-integrated spectral function for different interaction strengths. Top: total spectrum with respective Mn(3$d$) occupation $n_d$, Bottom: orbital-resolved projection onto the Mn(3$d$) states and the Si states. **d** Imaginary part of the Mn self-energy for different interaction strengths at small Matsubara frequencies $\omega_n = (2n + 1) \pi T$. Left: orbital-averaged –Im $\Sigma_{Mn}(i\omega_n)$ (symbols) with fitting function $F(\omega_n) = K\omega_n^\alpha$ (lines). Right: orbital-resolved self-energies for $U = 2$ eV and $J_H = 0.35, 0.65$ eV.

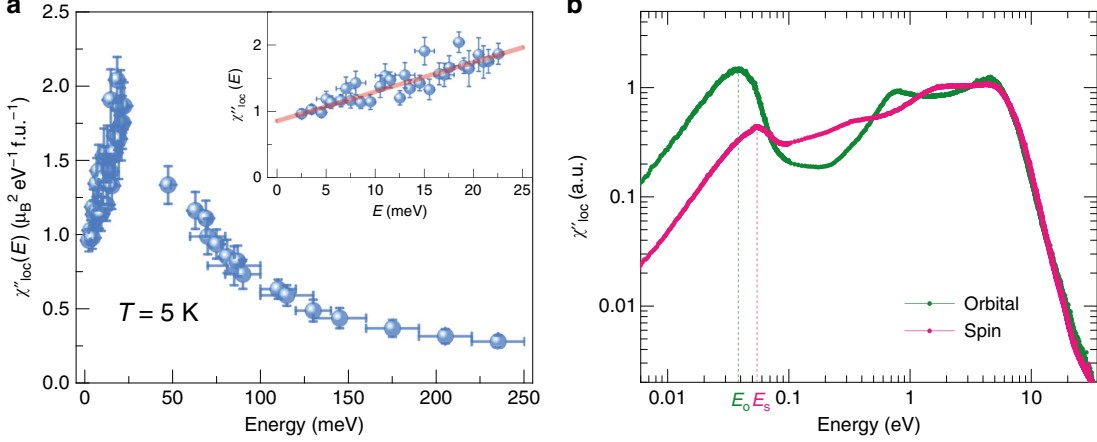

**Fig. 4 Local susceptibilities for MnSi. a** Local magnetic susceptibility determined via inelastic neutron scattering data plotted as a function of energy. The inset shows the low-energy region (below 25 meV) and the linear dependence well below the peak near 30 meV. **b** Local Mn susceptibility calculated via DFT+DMFT calculations for both orbital (green line) and spin (maroon line). Vertical dashed lines mark peaks above which coherence is lost in both the orbital and spin sectors. Error bars represent one standard deviation of the data.

(>300 meV), we ascribe broad peaks in the DFT+DMFT study to precoherence scales associated with e.g. short-range fluctuations about the Mn-ions. The remaining higher energy magnetic signal becomes increasingly diffuse and difficult to experimentally resolve above the background. Substantial spectral weight should however persist at these energies as predicted by the DFT+DMFT, and the integrated spectral weight from Fig. 4a is only $0.18 \pm 0.02 \ \mu_B^2$/f.u. and well below the paramagnetic local moment reported for MnSi[44].

The prominent role of orbital fluctuations is likely endemic to the MnSi crystal structure that provides a trigonal setting in a cubic symmetry. Within this unique environment, strong orbital fluctuations manifest that only become coherent at a comparatively low energy. Specifically, the orbital dependence in the scattering behavior has a dominant role here, as the orbital-resolved self-energies on the real-frequency axis (shown in Supplementary Fig. 8 and Note 6) reveal key splittings of the low-energy extrema within the imaginary part. This can explain the fact that fluctuations into the $a_{1g}$ orbital, which is designated by symmetry and proximity to Si, contribute the most to the emerging orbital coherence scale at low-energy. Figure 4 shows that the local susceptibilities are linear in frequency well below these coherence scales for both experiment and theory, in accordance with reaching a Fermi-liquid state[33]. Note that although the local susceptibilities are computed for $T = 300$ K, they still adequately represent the energy spectrum down to the Stoner-continuum onset. Experimentally, the temperature dependence in this energy range is very weak (Supplementary Fig. 5 and Note 3).

The loss of orbital coherence seemingly coincides with the divergence of the damped-magnon dispersion within the particle–hole continuum. This suggests the breakdown of coherent spin fluctuations due to the coupling of orbital and spin degrees of freedom as seen in the ferromagnetic Fermi-liquid theory[40,41]. Analysis of the DFT+DMFT orbital-resolved fluctuations (Supplementary Information) reveals that the coherence peaks are dominantly of itinerant $e_g'$ character in the spin channel as well as dominantly of $e_g - a_{1g}$ and $e_g' - a_{1g}$ character in the orbital channel. Hence in MnSi, a coupling of spin and orbital fluctuations via the $e_g'$ states seemingly drives the observed phenomenology. This provides an unconventional example of a Hund metal where first, the orbital coherence scale lies energetically below the spin coherence scale, and second, spin and orbital fluctuations are considerably intertwined.

Our findings suggest a much broader classification of $J_H$-driven correlated metals than previously considered. Importantly, many weak itinerant ferromagnets share many common aspects of the MnSi phenomenology. MnSi stands as a marked embodiment of $J_H$-driven effects (e.g. the size of the local moment, the magnitude of the mass renormalization and the size of the NFL regime), in line with its ideal Hund-metal local $d^6$ occupation of the transition-metal site. Other weak ferromagnets (e.g. ZrZn$_2$ or Ni$_3$Al) formed away from this ideal occupation likely still realize similar local correlation effects where electron counts larger or smaller than the ideal local occupation can carry over relevant Hund physics for a sizable $J_H/U$ ratio[47].

## Methods

**Inelastic neutron scattering experiments**. A single crystal with a mass of 52 g was used for time-of-flight inelastic neutron scattering experiments at SEQUOIA[48], Spallation Neutron Source at Oak Ridge National Laboratory. Data were collected with different incident energies $E_i$ = 18, 30, 70, 150, 300, 700 meV at $T$ = 5 K. Some higher temperature data (up to $T$ = 300 K) were also collected at select incident energies; however, no appreciable difference is found for high-energy spin wave excitations even at $T$ = 300 K. The background is corrected using the intensity at larger $Q$ zone centers where magnetic contributions are negligible. The magnetic form factor term for Mn$^{3+}$ in the scattering cross section is accounted for

and removed for after nonmagnetic background contributions were removed. Sample growth and characterization details can be found in ref. [49]. Data were analyzed using the software package Horace[50] as described in Supplementary Note 1.

**Ferromagnetic Fermi-liquid calculations**. We use the version of the ferromagnetic Fermi-liquid theory (FFLT), where the Fermi-liquid phenomenology is augmented with the Ginzburg–Landau theory for a small moment ferromagnet[40,41]. The transport and dynamical equations for the FFLT follow from the steady state and time evolution equations of the magnetization distribution function. This leads directly to the prediction of the existence of the amplitude mode when the $l = 1$ Fermi-liquid parameter is included we used other experimental measurements to get the magnitude of the amplitude mode for MnSi[40,41]. This is predicted to be about 50 meV at $q = 0$ and it drops to about 25 meV when the Goldstone mode crosses with the amplitude mode inside the particle–hole continuum. The spin hydrodynamic equations were developed here to make the plot of Fig. 2c as detailed in the Supplementary Note 2.

**DFT+DMFT calculations**. The cubic B20 crystal structure (space group P2$_1$3) of MnSi has an internal trigonal distortion along the <111> cube diagonal, which breaks inversion symmetry. As a result, there are seven Si neighbors for each Mn, grouped into three classes of different bond lengths (Supplementary Note 4). The primitive cell includes four formula units, with all Mn(Si) sites being symmetry-equivalent, and the Mn(3$d$) states split into three symmetry classes: $e_g$ (degeneracy 2), $e_{g'}$ (degeneracy 2) and $a_{1g}$ (degeneracy 1). A charge self-consistent combination of density functional theory (DFT) and dynamical mean-field theory (DMFT) is employed for the one-particle properties[51]. A mixed-basis pseudopotential method[52,53] is used for the DFT part with the generalized-gradient approximation. Within the mixed basis, localized functions for Mn(3$d$), Si(3$s$), (3$p$) and Si(3$d$) are utilized to reduce the plane-wave energy cutoff. The correlated subspace for the DMFT part consists of the effective Mn(3$d$) Wannier-like functions as obtained from a projected-local-orbital formalism[54]. A five-orbital Slater–Kanamori Hubbard Hamiltonian parametrized by a Hubbard $U$ and a Hund's exchange $J_H$ is applied on the Mn sites. Four symmetry-equivalent Mn sites are in the primitive MnSi cell with lattice parameter $a = 4.558$ Å[55,56]. The single-site DMFT impurity problem is solved by the continuous-time quantum Monte Carlo scheme in hybridization expansion as implemented in the TRIQS package[57,58]. A double-counting correction of fully localized-limit type is used[59]. The system temperature is set to $T = 300$ K. Paramagnetism is assumed in all the computations and spin–orbit coupling is neglected. One-particle spectral information is obtained from analytical continuation via the maximum-entropy method as well as the Pade method.

Experimental estimates of the strength of local Coulomb interactions in MnSi favor a rather low value of $U = 2$–3 eV[16], which guided the selection in our calculations. For reference, Fig. 3c shows a comparison of the $k$-integrated spectral function $A_{tot}(\omega)$ for $U = 2$ eV and $U = 4$ eV. Because of the large effective Mn bandwidth, there are no essential differences in $A_{tot}(\omega)$ between these different interaction strengths; however, the established renormalization of the strong original Mn(3$d$) peak at $-1.9$ eV within DFT toward the $-1.5$ eV observed in valence-band photoemission is better described for the lower $U = 2$ eV value. The site and orbital projection $A_{proj}(\omega)$ unravels the strong intermixing between especially Si(3$p$) and Mn(3$d$) states and shows that electronic correlations are effective in shifting the Si states into lower-energy regions. The remaining contributions close to $\varepsilon_F$ within the Mn manifold demonstrate that the Mn-$e_{g'}$ orbital supports the itinerant-electron character strongest.

Local susceptibilities are computed by DFT+DMFT without charge self-consistency using the Hirsch–Fye quantum Monte Carlo scheme. Analytical continuation of the local susceptibilities is performed with the stochastic optimization method (SOM)[60] (i.e. Mishchenko method) as implemented in the TRIQS/SOM package[61]. The standard SOM procedure is combined with projection on a very fine energy grid and subsequent rebinning onto a sparser grid. This approach strongly suppresses the stochastic noise and allows to reliably resolve the low-energy spectral features in the computed susceptibilities. The paramagnetic local magnetic moment $M$ on the Mn sites is obtained from computing $\chi_{loc}(\omega = 0)$ by integrating the local spin susceptibility over imaginary times, and employing the Curie–Weiss formula $\chi_{loc} = M^2/(3T)$.

## Data availability
All data are available from the corresponding authors upon request.

## Code availability
All codes used in generating results are available from the corresponding authors upon request.

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

## Acknowledgements

X.C. thanks fruitful discussions with Yi Zhang. F.L. thanks A. Georges and G. Kotliar for helpful discussions. K.S.B. thanks J. Heath, P. Farinas, K. Blagoev and Yi Zhang for insightful discussions. This work was supported by the MRSEC Program of the National Science Foundation under Award No. DMR 1720256 (S.D.W. and X.C.). F.L. is supported by the Deutsche Forschungsgemeinschaft (DFG) under the project LE-2446/4-1. I.K. is supported by The Simons Foundation. DFT+DMFT computations were

performed at the JUWELS Cluster of the Jülich Supercomputing Centre (JSC) under the project hhh08, as well as at the Physnet Computing Cluster of the University of Hamburg. D.R. was supported by the DOE, Office of Basic Energy Sciences, Office of Science, under Contract No. DE-SC0006939. A portion of this research used resources at the Spallation Neutron Source, a DOE Office of Science User Facility operated by Oak Ridge National Laboratory.

## Author contributions

T.W. synthesized and characterized the MnSi single crystals. X.C., D.R. and S.D.W. performed the inelastic neutron scattering experiment with the help from M.B.S. and A.I.K. X.C. analyzed the experimental data. S.D.W. and K.B. designed the experiments. F.L. and I.K. performed the DFT+DMFT calculations and analyzed the numerical data. X.C., F.L. and S.D.W. prepared the manuscript with the input from all other co-authors.

## Competing interests

The authors declare no competing interests.
