## [Peer Review File · Nature Communications]

Reviewers' comments:

Reviewer #1 (Remarks to the Author):

The manuscript by X. Chen (et al.) presents a study of high-energy magnetic excitations in MnSi well into the Stoner continuum. The spectra in this region consist of strongly damped single-particle excitations, as opposed to the collective spin-waves found at low momentum and energy transfers. They experimentally find that the damping parameter diverges before the boundary of the magnetic Brillouin zone, prematurely cutting off further magnon propagation. Using density functional theory and dynamical mean-field theory, the authors link this effect to an additional decay channel given by interorbital Hund's exchange.

The authors were able to collect high-quality experimental data of a region in the phase space of MnSi which is still very unexplored due to the difficulties that the measurements of such strongly damped and weak high-energy excitations pose. Together with their innovative theoretical interpretation, which establishes MnSi as an unconventional Hund metal, I recommend publication in Nature Communications once the following minor points have been addressed:

(1) As the study is specific to MnSi, it would be useful to include the concrete momentum and energy scales in Fig. 1(a).

(2) A DHO model is used to determine the line widths of the damped excitations that are depicted in Fig. 2(d). The instrumental resolution is not taken into account for these fits because the authors name an energy resolution of 1.6 meV for an incident neutron energy of 30 meV, i.e. ca. 10% of the damping constant. The instrumental resolution can nevertheless be expected to change significantly within the large range of energy transfers of 0..20 meV. Please ensure that it can still be neglected for up to 20 meV.

(3) Have all experiments been conducted at zero magnetic field? If so, does the zero-field multi-domain magnetic state of MnSi also play a role at high energies? In the low-energy helical state, for example, the presence of multiple domains leads to a complicated superposition of individual magnon modes, which the authors of Ref. 17 could only disentangle via resolution-deconvolution methods.

(4) To better distinguish the modified spectrum including the Hubbard and Hund terms from the pure non-magnetic DFT spectrum in Fig. 3, it might be useful to unite Fig. 3(a) and (b). For example, one could overplot the pure DFT lines of Fig. 3(a) in Fig. 3(b).

(5) In the same way as it is already done for Supplementary Fig. 5, it would be helpful to add temperature labels to all plots showing experimental data.

Reviewer #2 (Remarks to the Author):

This paper studied the nature of spin fluctuations in MnSi by analyzing the magnetic susceptibility data measured by inelastic neutron scattering (INS). The INS data reveals that the ferromagnetic spin-wave dispersion in MnSi becomes strongly damped inside the particle-hole continuum region. MnSi exhibits the strong coupling between the local magnetic moment from the Mn site and the itinerant bands in the Si site. Understanding the microscopic origin behind this physics is difficult due to the interplay between itinerant and localized nature of spin fluctuations. In this paper, DFT+DMFT is

employed to carry out the electronic structure and the local susceptibility calculations in MnSi. Theoretical calculations show that the nature of many-body correlations are local spin and orbital fluctuations captured in DFT+DMFT and the local susceptibility is peaked at the coherence energy which can be computed by DFT+DMFT. This material can be considered as a Hund's metal in which Hund's J plays an important role and the non-Fermi liquid behavior is also captured within DFT+DMFT. Overall, this paper is a nice collaboration paper between theory and experiment to reveal important features in spin fluctuations of the Hund's metal. While the data is technically sound and the arguments can be applicable to other similar materials, I think the following points should be clarified for the paper to be accepted in Nature Communications.

1. While the paper emphasizes the importance of spin and orbital fluctuations in capturing magnetic fluctuation features in MnSi, I am not fully convinced about the role of orbital correlations. Is the spin incoherence originated from the orbital incoherence given that the spin and orbital coherence energy scales are very close in DFT+DMFT calculations? Why are the interorbital fluctuations with the a_{1g} orbital dominant contributions?
2. What is the origin of additional bumps in the DMFT susceptibility above the coherence temperature? Are they originating from the fluctuations of the Hubbard bands? Or the Hund's coupling energy? Why is this feature not captured in the experimental susceptibility above the coherence temperature exhibiting the strong damping due to the particle-hole continuum?
3. How do the orbital and spin coherence energy scales change at different U and J values in DMFT?
4. The DFT+DMFT susceptibility was computed in the paramagnetic symmetry. Would the spin excitation and the coherence energy be still similar if it is computed using the ferromagnetic symmetry?
5. The authors also argue that correlations beyond the local one in ferromagnetic spin waves are important. At the same time, Fermi liquid theory calculations including $l=0$ and $l=1$ Landau parameters without the multi-band effect can still explain qualitatively the spin wave dispersion with a gapped mode as a function of momentum (Fig.2c). The authors should clarify the role of dynamical correlations on the spin wave dispersion. For example, how does the local DMFT susceptibility compare to the Lindhard susceptibility (without dynamical vertex correction) computing using the one-particle DMFT spectra or the DFT spectra?
6. In the main discussion, the authors argue that the DFT+DMFT magnetic moment is 0.3 μ_B while the experimental moment is 0.4 μ_B . Are those paramagnetic fluctuating moments or static ferromagnetic moments? It should be clarified in the text. How is this moment computed in DFT+DMFT? Is this moment consistent with the sum rule of the magnetic susceptibility data?
7. Can the authors show the orbital-resolved self-energies on the real axis? Does self-energy show any signature of coherence energy scale?
8. The authors claim that the Fermi surface is strongly modified due to correlations. Can the authors also plot the Fermi surface comparing DFT vs DFT+DMFT to show the correlation effect?
9. The authors say that local susceptibilities within DFT+DMFT without charge-self-consistency using the Hirsch-Fye QMC. Does this mean that they use DFT charge density or DMFT charge density? Also, why didn't they use CTQMC for computing the susceptibility? If there is any technical issue, they can clarify in the text.
10. The purple color in Fig.3.(a) is not visible even in the color print. The authors should use different

colors.

11. The orbital figures in Suppl. Fig. 6 are hidden by other atoms. The authors should display a smaller number of atoms to emphasize the shape of orbitals.

Reviewer #1 (Remarks to the Author):

The manuscript by X. Chen (et al.) presents a study of high-energy magnetic excitations in MnSi well into the Stoner continuum. The spectra in this region consist of strongly damped single-particle excitations, as opposed to the collective spin-waves found at low momentum and energy transfers. They experimentally find that the damping parameter diverges before the boundary of the magnetic Brillouin zone, prematurely cutting off further magnon propagation. Using density functional theory and dynamical mean-field theory, the authors link this effect to an additional decay channel given by interorbital Hund's exchange.

The authors were able to collect high-quality experimental data of a region in the phase space of MnSi which is still very unexplored due to the difficulties that the measurements of such strongly damped and weak high-energy excitations pose. Together with their innovative theoretical interpretation, which establishes MnSi as an unconventional Hund metal, I recommend publication in Nature Communications once the following minor points have been addressed:

We thank the referee for his/her careful review of our manuscript and subsequent recommendation for publication in Nature Communications. Below we detail changes to the manuscript and our replies to the referee's questions.

- (1) As the study is specific to MnSi, it would be useful to include the concrete momentum and energy scales in Fig. 1(a).**

This is a good idea. The revised Fig. 1(a) now contains this information.

- (2) A DHO model is used to determine the line widths of the damped excitations that are depicted in Fig. 2(d). The instrumental resolution is not taken into account for these fits because the authors name an energy resolution of 1.6 meV for an incident neutron energy of 30 meV, i.e. ca. 10% of the damping constant. The instrumental resolution can nevertheless be expected to change significantly within the large range of energy transfers of 0..20 meV. Please ensure that it can still be neglected for up to 20 meV.**

We followed the referee's suggestion and checked that the energy resolution can be safely ignored for all energy transfers between 0-20 meV. The instrument resolution narrows from 1.45 meV at 0 meV to 0.55 meV at 20 meV, and the damping constant continues to increase across this energy range as well. These values remain substantially smaller than the damping energy scale determined via our fits and the contribution from the spectrometer's intrinsic resolution can be safely ignored. The calculated resolution is shown here for reference.

(3) Have all experiments been conducted at zero magnetic field? If so, does the zero-field multi-domain magnetic state of MnSi also play a role at high energies? In the low-energy helical state, for example, the presence of multiple domains leads to a complicated superposition of individual magnon modes, which the authors of Ref. 17 could only disentangle via resolution-deconvolution methods.

Our experiments were indeed performed in zero field; however at the energies we are exploring in the present paper ($E > 3$ meV), we expect the influence of domains associated with the low energy helical state to be negligible. Namely, our data begin at 3 meV, which is the energy where magnons nominally enter the Stoner continuum. In this regime, there is a substantial, intrinsic damping that sets in that should supersede the effects of multidomain (anisotropic dispersion) broadening of modes endemic at low- q and low- E values. In the clean limit of a single domain, field polarized experiment, we know that above $q \sim 0.25 \text{ \AA}^{-1}$ spin waves already become intrinsically damped (Phys. Rev. B 16, 4956. (1977)). A nice way of further envisioning this is by referencing the data collected in Ishikawa et al. Journal of Applied Physics 49, 2125 (1978) where the spin spectrum was collected both in the field-polarized ordered state and in a high temperature, zero-field paramagnetic state. When comparing data at both extremes, there was minimal difference observed in the spin excitations within the p-h continuum above $\sim 3-4$ meV. This suggests that the high energy spectral weight focused on in our present paper is agnostic to the low energy, long-wavelength details of the ordered state and its potential domain structures.

It is possible that our lowest energy data point at 3 meV in Fig. 2d convolves some of the broadening due to helimagnons from the domains reported in Ref. 17 within the first reported γ value. We have now made a note of this in the revised text as well as provided a description of why this effect should not contribute to the remaining higher energy data in the paper.

(4) To better distinguish the modified spectrum including the Hubbard and Hund terms from the pure non-magnetic DFT spectrum in Fig. 3, it might be useful to unite Fig. 3(a) and (b). For example, one could overplot the pure DFT lines of Fig. 3(a) in Fig. 3(b)

We have now followed the referee's suggestion and overplotted the DFT lines with the DMFT calculation in a revised Fig. 3.

(5) In the same way as it is already done for Supplementary Fig. 5, it would be helpful to add temperature labels to all plots showing experimental data.

All plots in both the main text and supplemental information now show the temperatures that the data were collected at.

Reviewer #2 (Remarks to the Author):

This paper studied the nature of spin fluctuations in MnSi by analyzing the magnetic susceptibility data measured by inelastic neutron scattering (INS). The INS data reveals that the ferromagnetic spin-wave dispersion in MnSi becomes strongly damped inside the particle-hole continuum region. MnSi exhibits the strong coupling between the local magnetic moment from the Mn site and the itinerant bands in the Si site. Understanding the microscopic origin behind this physics is difficult due to the interplay between itinerant and localized nature of spin fluctuations. In this paper, DFT+DMFT is employed to carry out the electronic structure and the local susceptibility calculations in MnSi. Theoretical calculations show that the nature of many-body correlations are local spin and orbital fluctuations captured in DFT+DMFT and the local susceptibility is peaked at the coherence energy which can be computed by DFT+DMFT. This material can be considered as a Hund's metal in which Hund's

J plays an important role and the non-Fermi liquid behavior is also captured within DFT+DMFT. Overall, this paper is a nice collaboration paper between theory and experiment to reveal important features in spin fluctuations of the Hund's metal. While the data is technically sound and the arguments can be applicable to other similar materials, I think the following points should be clarified for the paper to be accepted in Nature Communications.

We thank the reviewer for his/her time and careful review of our manuscript. Below we address the referee's questions and provide the requested clarifications in the numbered points below.

1. While the paper emphasizes the importance of spin and orbital fluctuations in capturing magnetic fluctuation features in MnSi, I am not fully convinced about the role of orbital correlations. Is the spin incoherence originated from the orbital incoherence given that the spin and orbital coherence energy scales are very close in DFT+DMFT calculations? Why are the interorbital fluctuations with the a_{1g} orbital dominant contributions?

This is a good question, and indeed our conjecture is that the close behavior of the orbital and spin susceptibilities at low energy corroborates the message that, especially for this energy regime, there is coupling between the fluctuations. Aside from spin-orbit coupling, it is expected already intuitively that fluctuations in the respective orbital filling influences the spin fluctuations, simply because the spin-expectation value within an orbital connects to its filling. While this is an enticing framework to view things, the precise details mediating this apparent coupling take place at low-energy and an outstanding question at this point. Since this coupling is a low-energy feature, standard arguments on pure atomic(-like) mechanism are likely incorrect. On the other hand, standard Kondo-based arguments are difficult to apply to the manifest multi-orbital problem with the large degree of covalency of the MnSi compound. We

hope that our current findings inspire further work in this direction, importantly at the model level, in order to further clarify this conjecture.

Regarding the prominent role of a_{1g} fluctuations, our speculation is that the Mn-Si hybridization is crucial. This triggers the a_{1g} mode, since that very orbital is mostly linked to the Si neighbor(s) in a unique symmetry setting. Putting this into a more formal level, our self-energy analysis in the revised supplementary part (see below) demonstrates this contribution. On an intuitive level one may envision that the stronger coupling to Si marks that orbital as "the widest open door to the environment", thus "calming down" its fluctuations is seemingly the most difficult.

2. What is the origin of additional bumps in the DMFT susceptibility above the coherence temperature? Are they originating from the fluctuations of the Hubbard bands? Or the Hund's coupling energy? Why is this feature not captured in the experimental susceptibility above the coherence temperature exhibiting the strong damping due to the particle-hole continuum?

We attribute the "bumps" or "broad peaks" at energies between 100meV-1eV to the formation of pre-coherence of the orbital and spin fluctuations. This pre-coherence should be of more localized character, i.e. stemming from very short-range processes around the Mn sites, while the final coherence scales below 100meV are likely to connect more evidently to the intriguing low-energy physics of MnSi which takes place on a reasonably long time (Kondo) scale. Note that DMFT of course only describes local self-energies, however their self-consistent determination is obtained via the interplay of effective-atomic and bath effects. Features above the 1eV scale should be mainly associated with intra-atomic (short-time) excitations. Explicit Hubbard-band formation is rather weak in Hund metals, since the system is distant from a Mott-critical regime. Therefore, straightforward Hubbard excitations may not be crucial for the local susceptibilities. Regarding the role of Hund's coupling in the spectrum, while it is known from model calculations that the Hund coupling produces a side shoulder in the one-particle spectrum of 3-orbital Hund metals close to the main quasiparticle peak (Horvat et al., arXiv:1907:07100), it is not straightforward to provide the reason for an explicit Hund-related peak in the two-particle spectrum of local susceptibilities.

The fine features revealed in the model calculations are likely absent in the experimental data for two reasons: First, the multipeak structure at low energies expected for the combined orbital and spin susceptibilities falls within the range where we are unable to reliably disentangle phonon contamination of the spectrum. So while the data provides a sharp observation in reporting a low energy peak in the local spin susceptibility, resolving a multipeak structure in this energy range is problematic without future polarized neutron measurements. The second reason is that, at higher energies (>300 meV), the magnetic signal observed in experiment becomes increasingly diffuse and much more challenging to resolve fine structure. Kinematically, higher incident energy neutrons must be used in this regime, which yields poorer energy and momentum resolution and further weakens the magnetic signal via the larger momentum transfers required and diminished magnetic form factor. We have now added these points within the revised text.

That being said, we'd like to stress that, to the best of our knowledge, this is one of the first 5-orbital-based theoretical accounts providing the local susceptibilities for a realistic material with spectral features identified such a high energy resolution. The complexities apparent in the model from such 5-orbital spectra render it an open challenge to assign a sharp identification of all the newly observable features provided by our model, and we hope our results motivate future work exploring this fascinating new insight into this and related materials.

3. How do the orbital and spin coherence energy scales change at different U and J values in DMFT?

We have now provided this comparison by adding into the supplementary part of the revised manuscript an additional plot, comparing the local susceptibilities at $U=2$ eV for different Hund J_H . We observe that the smaller $J_H=0.35$ eV shifts the bumps in the intermediate energy range of the local susceptibility to somewhat higher energies and tends to align them. This is an effect expected for a conventional Hund metal, where a larger Hund contribution should split spin and orbital coherence scales. Yet the coherence scales at lower energy below 100meV are further split apart by a smaller Hund exchange. This marks the unconventional character of the MnSi Hund metal.

The non-charge self-consistent Hirsch-Fye calculations for $U=4$ eV yield a Mn(3d) filling larger than for the charge self-consistent CT-QMC calculations (i.e. closer to $n=7$). Therefore, we did not include further susceptibility data for that larger U value. However, one would expect to observe the trends for smaller J_H to be continued for larger U.

4. The DFT+DMFT susceptibility was computed in the paramagnetic symmetry. Would the spin excitation and the coherence energy be still similar if it is computed using the ferromagnetic symmetry?

This is a good question. The ferromagnetic (helical) ordering of course happens at much lower temperatures where spin coherence is seemingly achieved. It is expected that in this broken-symmetry state, that the fluctuations may not cover all phase space. However, the longitudinal fluctuations, which are mainly responsible for the local-moment formation are not that dependent on the magnetic ordering (the low energy transverse ones outside the continuum at $E < 3$ meV should matter more in that regime). Therefore, we would not expect major qualitative changes to the established picture we present at high energies.

5. The authors also argue that correlations beyond the local one in ferromagnetic spin waves are important. At the same time, Fermi liquid theory calculations including $l=0$ and $l=1$ Landau parameters without the multi-band effect can still explain qualitatively the spin wave dispersion with a gapped mode as a function of momentum (Fig.2c). The authors should clarify the role of dynamical correlations on the spin wave dispersion. For example, how does the local DMFT susceptibility compare to the Lindhard susceptibility (without dynamical vertex correction) computing using the one-particle DMFT spectra or the DFT spectra?

To address the referee's question, in the revised manuscript, we have added a comparison of the obtained DFT+DMFT χ_{loc} spin spectrum including vertex contributions (i.e. a manifest two-particle object) with a so-called "bubble" or "RPA(-like)" calculation based on (convoluted) one-particle Green's functions. This comparison shows that while there is some structure in the intermediate energy range without vertex corrections, the low-energy coherence scales are completely absent. Hence, the latter features truly ask for a faithful inclusion of vertex corrections. We have clarified this in the main text.

6. In the main discussion, the authors argue that the DFT+DMFT magnetic moment is 0.3 μ_B while the experimental moment is 0.4 μ_B . Are those paramagnetic fluctuating moments or static ferromagnetic moments? It should be clarified in the text. How is this moment computed in DFT+DMFT? Is this moment consistent with the sum rule of the magnetic susceptibility data?

The 0.3 μ_B value is based on paramagnetic fluctuating moments at $T=300K$, and is straightforwardly computed from using the Curie-Weiss formula relating the local moment M to the static susceptibility χ_{loc} via $\chi_{loc}=M^2/(3T)$, as usually done in DFT+DMFT calculations. The quantity $\chi_{loc}=\chi_{loc}(\omega=0)$ is computed from an integration of $\chi_{loc}(\tau)$. The experimental moment originally quoted was obtained from the ordered low-T phase; however we now realize that a direct comparison of these two values may be confusing for the reader. To address this, we have now moved text previously in the methods section into the main text, which details a more in-depth comparison between the DMFT+DFT paramagnetic moment value with the Curie-Weiss local moment value observed in experiment. The latter increases with temperature due to nonlocal spin-spin correlations as described by the SCR theory. This effect is not faithfully included in DMFT accounting for the difference, and an additional comparison with the instantaneous local moment is also made.

A trivial sum rule is difficult to apply to the experimental data due to the highly itinerant nature of the material, nevertheless one can use this as a consistency check. The experimental sum of the measured inelastic spectral weight is only 0.18 μ_B^2 . Given the large local moment of 2.2 μ_B seen in experiment, this implies there is substantial spectral inelastic weight at higher energies not captured in our experimental data (for the reasons discussed above) and consistent with the DFT+DMFT predictions. We have made these points in the revised text.

7. Can the authors show the orbital-resolved self-energies on the real axis? Does self-energy show any signature of coherence energy scale?

This is indeed a good suggestion and we have included a plot of the orbital-resolved self-energies in the revised supplementary section. Therefrom, we were able to identify two different maxima in the imaginary part of the self-energy on the real axis at low energy. This may explain the orbital peak below 100meV, and especially its dominant character from fluctuations into the a1g orbital. We have added reference to this result in the main text.

8. The authors claim that the Fermi surface is strongly modified due to correlations. Can the authors also plot the Fermi surface comparing DFT vs DFT+DMFT to show the correlation effect?

We have now added a new figure in the supplementary part of the revised manuscript that compares the DFT and the DFT+DMFT Fermi surface in a chosen two-dimensional cut. There are clear differences, as previously noted in the main text.

9. The authors say that local susceptibilities within DFT+DMFT without charge-self-consistency using the Hirsch-Fye QMC. Does this mean that they use DFT charge density or DMFT charge density? Also, why didn't they use CTQMC for computing the susceptibility? If there is any technical issue, they can clarify in the text.

There is no technical issue, but rather a coding issue. Our homegrown DFT+DMFT code uses the triqs package for the DMFT impurity solution in an earlier version. In that version, computation of local susceptibilities is not yet implemented. Thus, we used our Hirsch-Fye code, which is yet not available in charge self-consistent mode, to compute the local susceptibilities. We checked that our physical description is robust independent of the specific DMFT flavor. The Hirsch-Fye calculations have been performed in the standard 'one-shot' architecture, i.e. employing DMFT self-consistency by starting from the DFT charge density.

10. The purple color in Fig.3.(a) is not visible even in the color print. The authors should use different colors.

We have now updated Fig. 3(a) with a different color.

11. The orbital figures in Suppl. Fig. 6 are hidden by other atoms. The authors should display a smaller number of atoms to emphasize the shape of orbitals.

We have now revised the Suppl. Fig. 6 accordingly.

REVIEWERS' COMMENTS:

Reviewer #1 (Remarks to the Author):

With their revised manuscript submission, the authors have answered all my questions and added the suggested changes. Therefore, I fully support the publication of this important work in Nature Communications.

Reviewer #2 (Remarks to the Author):

I can see that the authors addressed my questions in the proper manner. Therefore, I recommend the manuscript for the publication now.